# Safe abortion service utilization and associated factors among insecurely housed women who experienced abortion in southwest Ethiopia, 2021: A community-based cross-sectional study

**Kidist Alemu**[1], **Solomon Birhanu**[2], **Leta Fekadu**[2], **Fitsum Endale**[3], **Aiggan Tamene**[3], **Aklilu Habte**[3]*

1 Mizan Aman College of Health Science, Southern Region Health Bureau, Mizan, Ethiopia, 2 Department of Epidemiology, Faculty of Public Health, Institute of Health, Jimma University, Jimma, Ethiopia, 3 School of Public Health, College of Medicine and Health Sciences, Wachemo University, Hosanna, Ethiopia

* akliluhabte57@gmail.com

## Abstract

### Background

Insecurely housed women are more vulnerable to physical and mental health issues than the general population, making access to a safe abortion more difficult. Though Ethiopia has a penal code regarding safe abortion care, there has been a dearth of studies investigating the safe abortion care practice among those insecurely housed women. Thus, this study aimed at assessing the magnitude of safe abortion service uptake and its determinants among insecurely housed women who experienced abortion in southwest Ethiopia.

### Methods

A community-based cross-sectional study was conducted in three towns in southwest Ethiopia from May 20-July 20, 2021. A total of 124 street-involved women were included in the study. They were selected by snowball sampling technique and data was collected through a face-to-face interview. The data were entered into Epi-data Version 3.1 and exported to SPSS 21 for analysis. A bivariable and multivariable logistic regression analyses were performed to determine the association of independent variables with the outcome variable. The level of significance was determined at a p-value <0.05. To determine whether the model is powerful enough in identifying any significant effects that do exist on the dependent variables, a power analysis was performed via a Post-hoc Statistical Power Calculator for Multiple Regressions.

### Results

The magnitude of safe abortion service utilization among insecurely housed women was found to be 27.9% [95% CI: 20.1, 34.2]. Average daily income [AOR:3.83, 95% CI: 1.38,

**Data Availability Statement:** All relevant data are within the article and its Supporting Information files.

**Funding:** The author(s) received no specific funding for this work.

**Competing interests:** The authors have declared that no competing interests exist.

**Abbreviations:** AOR, adjusted odds ratio; COR, crude odds ratio; EDHS, Ethiopia demographic and health survey; HCPs, Health care providers; WHO, World Health Organization.

10.60], knowledge of safe abortion services [AOR:3.94; 95% CI: 1.27,9.24], and affordability of the service [AOR: 3.27; 95% CI:1.87, 8.41] were identified as significant predictors of safe abortion service among insecurely housed women.

## Conclusion and recommendation

The magnitude of safe abortion service utilization among insecurely housed women in the study area was low. The respective town health offices and health care providers at the facility level should strive to improve awareness about safe abortion service's legal framework, and its availability. In addition, a concerted effort is needed from local administrators, NGOs, and healthcare managers to engage those insecurely housed women in income-generating activities that allow them to access safe abortion and other reproductive and maternal health services.

## Introduction

According to the World Health Organization (WHO), abortion is considered safe when performed using a method that is appropriate for the gestational age and by trained health professionals in a safe and clean environment [1, 2]. Unintended pregnancy as a result of unmet contraceptive needs and limited access to safe and legal abortion are two major reasons that contribute to unsafe abortion [3]. Insecurely housed women, or those who do not have a definite, regular, and suitable place to sleep at night, are the most vulnerable group to unwanted pregnancies [4]. This is mainly because they are subjected to sexual violence, rape or harassment, and a low level of contraceptive utilization [5].

Despite their clear desire to avoid pregnancy, Insecurely housed women face hurdles to accessing reproductive health care [6]. Furthermore, the majority are unaware of the availability of safe abortion care options, and the fear of stigma and discrimination prevents them from accessing these services [7]. As a result, they frequently exposed unsafe abortions, such as self-induced abortions, to avoid the perceived shame and stigma associated with seeking abortion care [8].

Unsafe abortion is one of the leading causes of maternal mortality, accounting for 4.7 to 13.2 percent of all maternal deaths worldwide, leaving 220,000 children without a mother each year [9]. According to data, 30 women die per 100,000 unsafe abortions in developed countries, while 220 women die per 100,000 unsafe abortions in developing countries annually. In addition, over 7 million women are treated in health facilities each year for consequences of unsafe abortion, including bleeding and sepsis [1, 10]. As a result, having access to a safe and high-quality abortion service is crucial in reducing the number of unsafe abortions and the accompanying maternal deaths and complications [2].

Between 2015 and 2019, 73.3 million induced (safe and unsafe) abortions were performed annually on a worldwide scale. Induced abortions occurred in 3 out of 10 (29%) of all pregnancies and 6 out of 10 (61%) of all unwanted pregnancies and among these, 1 out of 3 was carried out in the least safe or dangerous conditions [11]. Unsafe abortion is responsible for between 4.7–13.2% of maternal deaths per year [10]. According to estimates from 2010 to 2014, over 45% of all abortions were unsafe, with nearly all of these unsafe abortions occurring in developing countries [12]. The highest incidences of unsafe abortion occur in Latin America, Africa, and Southeast Asia [13]. In Africa, the chance of dying as a result of unsafe abortion was the highest [12].

Abortion has been legalized in Ethiopia since 2005 in cases of rape, incest, fetal impairment posing a life-threatening risk to the mother or her child, or if the mother is unable to raise her

child due to physical or mental incapacity. Despite the law, many Ethiopian women have diffi-culties accessing safe abortion [9, 14]. In 2014, the national estimate of induced abortion was 620,300, with an annual abortion rate of 28 per 1,000 reproductive-age women [15]. Despite advances in the availability of safe abortion services, about half of all abortions continue to take place outside of health facilities [16].

As compared to the general population, those pregnant women who are insecurely housed are more vulnerable to physical and mental health issues that affect both the mother and the baby, and access to a safe abortion becomes more difficult [17]. Furthermore, insecurely housed women seek abortion care later in pregnancy, leading to a higher likelihood of abortion complications [18]. In general, due to limited access to routine health care, the risks of all preg-nancies are increased in those groups [19].

According to studies, the use of safe abortion services by insecurely housed women is quite low. Only 19% and 10.4% of them in the United States and Brazil, respectively, accessed safe abortion services [18, 20]. In the Democratic Republic of Congo (DRC), 24.5% of prior preg-nancies resulted in induced abortion, with the proportion among the youngest street girls (50%) being the highest [21]. A study on the reproductive health of insecurely housed women in Addis Ababa found that almost all (96%) of pregnancies were unwanted, with 59.4% of these pregnancies ending in abortion [22]. Safe abortion services should be easily accessible and affordable to all women, according to the WHO [2]. However, evidence suggests that even in countries with liberal abortion laws, impediments to safe abortion, such as legal, health-care, socioeconomic, and stigma-related barriers, hamper access to safe abortion services [23].

Women who are insecurely housed are usually underrepresented. In Ethiopia, the number of insecurely housed women is rising rapidly, yet their problems are not being addressed in government plans [24]. Furthermore, just a few studies on their reproductive health issues have been conducted in Ethiopia [22, 25, 26]. Even though insecurely housed women have a higher risk of unwanted pregnancies than the general population, the majority of studies have focused on the use of safe abortion services by the general populace and students [27–29]. This study is therefore important in identifying the level of and factors related to safe abortion ser-vice utilization, and also providing information that will aid in addressing the reproductive health needs of insecurely housed women in the study area.

## Methods and material

From May 20 to July 20, 2021, a community-based crosssectional study was conducted in three purposively selected towns in southwest Ethiopia: Jimma, Bonga, and Mizan-Aman. Jimma town is the capital of the Jimma zone in the Oromia region of Ethiopia, which is located 345 km from Addis Ababa, the capital city of Ethiopia. The town has 17 kebeles (13 urban & 4 rural) and based on the 2007 national census, the estimated population of Jimma town in 2020/21 is 220,609. Bonga town, the capital of the Kaffa zone, is 105 kilometers and 465 kilo-meters away from Jimma town and Addis Ababa, respectively. The town has three administra-tive kebeles and a total population of 27,634 people, with 13624 men (49.3%) and 14010 women (50.7%). Mizan-Aman is the capital of the Bench-Sheko zone and is located 651 kilo-meters southwest of Addis Ababa. The total population of the town is estimated to be 48,934, with 24,956 women (51%) living in all five kebeles.

### Populations of the study

All insecurely housed women with abortion experience living in three selected towns of south-west Ethiopia (Jimma, Bonga, and Mizan Aman) were the source population. The study popu-lations were those selected insecurely housed women with abortion experience residing in the

towns. All insecurely housed women with a previous history of abortion were eligible to take part in this study. Participants who were seriously ill during the data collection period were excluded from the study.

## Sample size determination

The sample size for the study was determined using the rule of thumb of 10 participants per measurement variable [30–32]. The number of explanatory variables after an in-depth review of the literature was 11, and thus the minimum sample size required for this study was (10 x 11 = 110). After accounting for a 10% non-response rate, the final sample size was 124.

## Sampling technique and procedures

A snowball technique was used to choose study participants. The data was collected from the first group of eligible insecurely housed women who experienced abortion and agreed to participate in the study. These women were then used as a reference for recruiting another respondent who could be included in the study. The data collection proceeded until a large number of women had been identified and the required sample size had been met.

## Data collection tool, methods, and personnel

Data were collected using a pre-tested structured questionnaire that was prepared after reviewing relevant literature in trying to attain the study's objectives [1, 2, 20, 24]. The tool had six sections: Background characteristics, obstetric characteristics (pregnancy and abortion experiences), individual factors (knowledge of safe abortion services), health system-related characteristics, and utilization of safe abortion services. Four BSc midwives collected data through face-to-face interviews under the supervision of three public health officers. Before data collection, the respondents were informed of the study's purpose and asked if they were willing to take part in it.

## Data quality management

The questionnaire was translated from English to Amharic, and then English language instructors will translate it back into English. Cross-cultural and conceptual translations were prioritized during the translation process over terminological literality or linguistic equivalence. Furthermore, the questionnaire was pre-tested on 5% of the total sample size in Serbo town, outside the study area. The internal consistency between the items in the knowledge and practice assessment questions was evaluated, and the corresponding Cronbach's alpha values were 0.83 and 0.87, respectively. The overall internal consistency composite score was 0.85, which indicates that 85% of the variance in the scores was found to be reliable and deemed to be good. Both data collectors and supervisors got a one-day intensive training on the objectives of the study, data collection techniques, and procedures. Finally, supervisors reviewed and checked questionnaires for completeness at the end of each data collection day, and appropriate feedback was given to data collectors the very next day. The privacy of study participants was maintained to encourage genuine participation and information sharing as sources for the study.

## Measurement of variables of the study

The outcome variable was the use of safe abortion services. Those who have terminated their pregnancy in a health facility by a certified health care provider were considered to have had a safe abortion (YES = 1), while those who sought abortion outside of a health facility were considered to have had an unsafe abortion (NO = 0).

Insecurely housed (street) women: Women who live or spend most of their time on the street and rely on it for their livelihood [33, 34].

On-the-street women: were those women who had no formal homes (insecurely housed) and sleep on streets, verandas, balconies, etc. at night [34, 35].

Off-the-street women: are those women who have houses to go to for sleep at night while making their lives on the street life [34, 35].

Knowledge of abortion service: Eight questions were used to assess respondents' level of knowledge of abortion services, and those who scored at or above the mean were considered knowledgeable unless they were not [36].

The perceived waiting time for abortion services: is the average time from initial referral to procedure reception, and it is considered prolonged if it exceeds 2 hours and short if it is less than 2 hours.

## Data analysis

The data were entered into EpiData version 3.1 and exported to SPSS version 21 For further analysis. Inconsistencies, completeness, and outliers were checked using running frequency, cross-tabulations, and sorting. To check for the distribution of variables throughout the population, descriptive statistics such as frequency distribution, proportion, mean, and standard deviation were computed. A binary logistic regression analysis was used to find factors associated with safe abortion service utilization. A bivariable logistic regression was used to assess the relationship between each explanatory variable and the response variable. As a result, with a P-value <0.25, eight of the eleven variables showed an association and were entered into a multivariable logistic regression model. Finally, multivariable logistic regression was performed, and three statistically significant variables at p-value <0.05 were identified. The adjusted odds ratio (AOR) of each significant variable with the corresponding 95 percent confidence interval was used to report the strength and direction of the association.

To determine whether the model is powerful enough to identify any significant effects that do exist on the dependent variables, a Post-hoc power analysis was performed by using a Post-hoc Statistical Power Calculator for Multiple Regression. The following parameters were considered during the calculation: the number of predictors in the final model = 3, observed $R^2$ = 0.28, probability level = 0.05, and a sample size = 122. Finally, the observed statistical power was found to be 0,9998, which is much greater than the minimum requirement of 0.8. So, enough statistical power existed to detect a significant effect.

To assess model fitness, the effect size of the final model was estimated by correlating the predictive probabilities of each case (explanatory variables) with the outcome of interest. By doing so, the correlation value(the predicted probability) was 0.575, possibly an indicator of a good model fit. Furthermore, model fitness was assessed using the Nagelkerke R Square and the Omnibus test, the results of which were 0.466 and 0.001, respectively, indicating a good model fit. Multicollinearity between independent variables was also checked by estimating the variance inflation factor and no multicollinearity was detected (VIF<10).

## Ethical consideration and consent to participate

Ethical clearance was obtained from Jimma University Health Institute's Human Research Ethics Committee. A letter of cooperation was also obtained from the Epidemiology department for the health offices of three selected towns. A letter of permission was obtained from each health office to proceed with the study. The participants were informed about the purpose of the study, and their right to refuse participation and discontinue the interview. Written Informed consent was obtained from each participant before the interview. By avoiding any

identifiers of the study participants, the information obtained from them was kept confidential throughout the study. Personal protective equipment was provided for data collectors because the study takes place amid a global pandemic, COVID-19.

## Results

### Socio-demographic characteristics of the respondents

A total of 122 insecurely housed women with abortion experience were interviewed, yielding a response rate of 96.0%. Regarding the place of residence, 96(78.7%), 17(13.9%), and 9(7.4%) were from Jimma, Bonga, and Mizan-Aman towns, respectively. the majority, 90 (73.8%) of the study participants were "off-the-street". The mean(±SD) age of respondents was 26.14 (±6.71) years. Seventy-one (58.2%) of respondents had ever been married, and more than half (51.6%) had no formal education. The average(±SD) daily income of the respondents was 21.35(±6.79) ETB. The majority of respondents (84.4%) relied primarily on panhandling ("begging") for a living, with the remaining 19 (15.6%) working part-time jobs in addition to panhandling (Table 1).

**Table 1. Socio-demographic information of insecurely housed women in southwest Ethiopia, 2021.**

| Variables | Safe abortion | | Total(%) |
|---|---|---|---|
| | Yes(%) | No(%) | |
| **Age category in years** | | | |
| <25 | 18 (31.1) | 40 (68.9) | 58(47.5) |
| ≥25 | 16 (25.0) | 48 (75.0) | 64(52.5) |
| **Marital status** | | | |
| Un married | 15 (29.4) | 36 (70.6) | 51(41.8) |
| Ever married | 19 (26.7) | 52 (73.2) | 71(58.2) |
| **Religion** | | | |
| Orthodox | 13 (25.5) | 38 (74.5) | 51(41.8) |
| Muslim | 11 (28.2) | 28 (71.8) | 39(32.0) |
| Protestant | 10 (31.2) | 22 (68.8) | 32(26.2) |
| **Ethncity** | | | |
| Oromo | 14(35.0) | 28 (65.0) | 42(34.4) |
| Dawro | 7 (21.2) | 26 (78.8) | 33(27.0) |
| Amhara | 9(30.0) | 21 (70.0) | 30(24.6) |
| Yem | 2(28.6) | 10(71.4) | 12(9.8) |
| Others | 2(40.0) | 3 (60.0) | 5(4.1) |
| **Educational level** | | | |
| No formal education | 13 (20.6) | 50 (79.4) | 63(51.6) |
| Primary school | 21 (35.6) | 38(64.4) | 59(48.4) |
| **Insecurely housed women by type** | | | |
| Off-the-street | 25(27.8) | 65(72.2) | 90(73.8) |
| On-the-street | 9(28.1) | 23(71.8) | 32(26.2) |
| **Daily income** | | | |
| Below average (<21.35ETB) | 16(19.3) | 67(80.7) | 83(68.0) |
| Average and above(≥21.35ETB) | 18(46.2) | 21(53.8) | 39(32.0) |
| Source of income | | | |
| Begging (Panhandling) | 27(26.2) | 76(73.8) | 103(84.4) |
| Begging+other work | 7(36.4) | 12(63.2) | 19(15.6) |

## Individual and health system-related characteristics

Nearly three-fourths, 90 (73.8%) of respondents were not knowledgeable about abortion. Regarding stigma, 37(30.3%) of respondents reported that they have ever faced stigma/discrimination from health care providers. Thirty-five(28.7%) respondents reported that they had been counseled at a health facility to continue the pregnancy. More than one-third, 43(35.2%) of study participants responded that they have waited a long time at a health facility and 41 (33.6%) reported that the cost of abortion services at a health facility was affordable to them (Table 2).

## Safe abortion service utilization

Thirty-four, 27.9% [95% CI: 20.1, 34.2] of insecurely housed women in the study area got a Safe abortion service. The majority(72.1%) of total abortion takes place outside health institutions under unsafe conditions. The commonest health facility visited by 15(44.1%) of safe abortion users were health centers followed by private clinics, 10 (29.4%), and public hospitals, 9(26.5%) (Fig 1). Of the 88 women who seek abortion outside of a health facility, 29 (32.9%) and 59 (67.1%) initiate abortions on their own (self-induced) and in an informal setting by a traditional provider, respectively. Fifty-two (42.6%) respondents said they knew women who had had abortions, while the remaining 72(57.4%) said they didn't.

## Determinants of safe abortion service utilization

In a multivariate logistic regression analysis, three variables were found to be significantly associated with safe abortion service utilization among insecurely housed women: knowledge of safe abortion care, average daily income, and affordability of safe abortion services at the health facility. Women with at least a daily average income had 3.83 times more chances of

**Table 2. Individual and health system-related characteristics of insecurely housed women in southwest Ethiopia, 2021 G.C.**

| Variables | Safe abortion | | Total (%) |
|---|---|---|---|
| | Yes(%) | No(%) | |
| **Perceived stigma and discrimination from HCPs** | | | |
| Yes | 15(22.3) | 52(77.7) | 67(54.9) |
| No | 19(34.5) | 36(65.5) | 55(45.1) |
| **Ever faced stigma by HCPs** | | | |
| Yes | 7(18.9) | 30(81.1) | 37(30.3) |
| No | 27(31.8) | 58(68.2) | 85(69.7) |
| **Knowledge of safe abortion services** | | | |
| Poor | 18(20.0) | 72(80.0) | 90(73.8) |
| Good | 16(50.0) | 16(50.0) | 32(26.2) |
| **Counseled to continue the pregnancy** | | | |
| No | 22(25.3) | 65(74.7) | 87(71.3) |
| Yes | 12(34.3) | 23(65.7) | 35(28.7) |
| **Perceived time for abortion care** | | | |
| Long (>1hr) | 9(20.9) | 34(79.1) | 43(35.2) |
| Short(<1hr) | 25(31.6) | 54(68.4) | 79(64.8) |
| **Affordability of cost for abortion services** | | | |
| Unaffordable to me | 17(21.0) | 64(79.0) | 81(66.4) |
| Affordable to me | 17(41.5) | 24(58.5) | 41(33.6) |

## The Place where safe abortions takes place

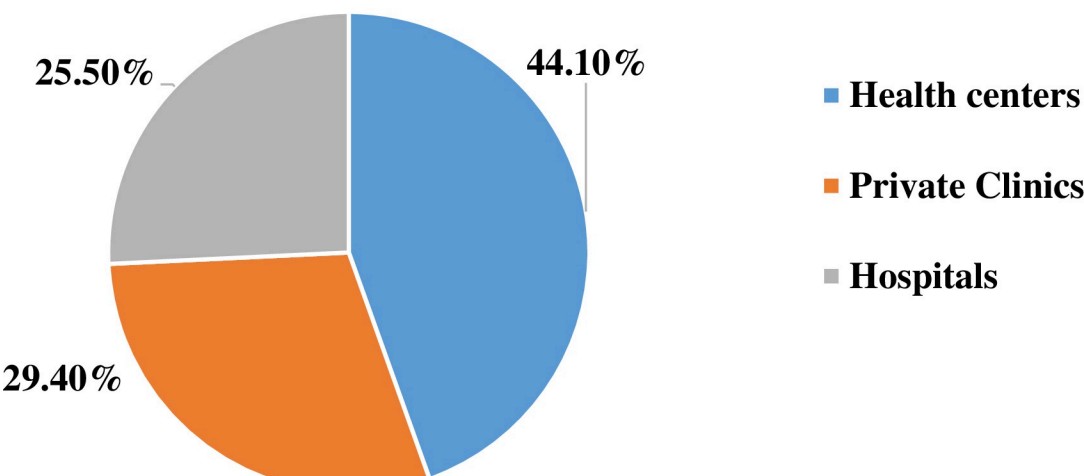

**Fig 1. The list of health facilities visited by insecurely housed women for safe abortion service in southwest Ethiopia, 2021.**

having a safe abortion than those with a daily income below the average [AOR:3.83, 95% CI: 1.38, 10.60]. Those insecurely housed women who had good knowledge of safe abortion services were 3.94 times more likely to receive one than those who had poor knowledge of safe abortion services [AOR:3.94; 95% CI: 1.27,9.24]. The odds of safe abortion service uptake were 3.27 times higher among women who reported that the service is affordable to them versus women who reported that the service is not affordable to them [AOR: 3.27; 95% CI:1.87, 8.41] (Table 3).

## Discussion

The purpose of this study was to estimate the magnitude and factors that influence the use of safe abortion services among insecurely housed women in Southwest Ethiopia. According to the findings, just 27.9% of insecurely housed women in the study area had a safe abortion, implying that nearly three-quarters of abortions among insecurely housed women were unsafe. This figure is significantly lower than the national estimate of safe abortion in 2014, which indicated that 53% of abortions were performed in health institutions under safe conditions [15]. The finding is also lower than those studies conducted in Addis Ababa (59.4%) and Gondar(55.5%) [22, 25]. The disparity could be owing to variation in the study subjects, as this study focused on insecurely housed women of reproductive age, whereas the studies in Addis Ababa and Gondar focused on street children aged 10–18 and 10–24 years old, respectively [22, 25]. This suggests that a higher number of insecurely housed women terminate their pregnancies outside of health institutions, potentially exposing them to increased risks of morbidity and mortality from unsafe abortion complications.

Safe abortion service utilization among insecurely housed women was significantly associated with variables such as knowledge of safe abortion care, average daily income, and affordability of safe abortion services at a health facility.

Insecurely housed women with higher average daily incomes were more likely to utilize safe abortion care. This finding is consistent with studies conducted in Chili and Northern Ethiopia, indicating that women with higher average daily incomes were more likely to utilize

**Table 3. Factors associated with safe abortion service utilization among insecurely housed women in southwest Ethiopia, 2021.**

| Variables | Safe abortion care | | COR (95% CI) | AOR(95% CI) | P-value |
|---|---|---|---|---|---|
| | **Yes** | **No** | | | |
| **Educational level** | | | | | |
| No formal education | 13 (20.6) | 50 (79.4) | 1 | 1 | |
| Primary school | 21 (35.6) | 38(64.4) | 2.13(0.94,4.78)* | 1.72(0.64,4.59) | 0.282 |
| **Age category** | | | | | |
| <25 | 18 (31.1) | 40 (68.9) | 1 | | |
| ≥25 | 16 (25.0) | 48 (75.0) | 1.35(0.61,2.98) | | |
| **Marital status** | | | | | |
| Un married | 15 (29.4) | 36 (70.6) | 1 | | |
| Ever married | 19 (26.7) | 52 (73.2) | 1.14(0.51,2.54) | | |
| **Daily income** | | | | | |
| Below average | 14(16.9) | 69(83.1) | 1 | 1 | |
| Average and above | 20(51.3) | 19(48.7) | 3.59(1.56,8.26)* | **3.83(1.38,10.60)**** | **0.010** |
| **Counseled to continue the pregnancy** | | | | | |
| No | 22(25.3) | 65(74.7) | 1 | | |
| Yes | 12(34.3) | 23(65.7) | 1.54(0.66.3.60) | | |
| **Knowledge of the safe abortion** | | | | | |
| Poor | 18(20.0) | 72(80.0) | 1 | 1 | |
| Good | 16(50.0) | 16(50.0) | 4.00(1.68,9.49)* | **3.94(1.27,9.24)**** | **0.018** |
| **Affordability of safe abortion service** | | | | | |
| Unaffordable for me | 14(17.3) | 67(82.7) | 1 | 1 | |
| Affordable for me | 20(48.8) | 21(51.2) | 2.67(1.17,6.05)* | **3.27(1.87.8.41)**** | **0.034** |
| **Perceived time for abortion care** | | | | | |
| Long (>1hr) | 9(20.9) | 34(79.1) | 1 | 1 | |
| Short(<1hr) | 25(31.6) | 54(68.4) | 1.75(0.73,4.19)* | 2.35(0.79,6.06) | 0.073 |
| **Perceived stigma and discrimination from HCPs** | | | | | |
| Yes | 15(22.3) | 52(77.7) | 1 | 1 | |
| No | 19(34.5) | 36(65.5) | 1.83(0.82,4.07)* | 1.68(0.65,4.38) | 0.274 |
| **Ever faced stigma by HCPs** | | | | | |
| Yes | 7(18.9) | 30(81.1) | 1 | 1 | |
| No | 27(31.8) | 58(68.2) | 1.99(0.78,5.11)* | 1.83(0.61,4.50) | 0.130 |

1: Reference category, AOR:Adjusted Odds Ratio,COR: Crude Odds Ratio

* significant at p-value< 0.25

** significant at p-value< 0.05

safe abortion [7, 37, 38]. Even though abortion was given freely in a semi-liberal approach in Ethiopia, insecurely housed women with low income may have been unable to access service delivery points because of some additional costs of transportation and medication, leading them to undergo unsafe abortion services [39]. Women with low incomes, on the other hand, may face challenges related to the fate of their newborns while growing up in a fatherless environment and may be afraid to raise their newborns independently in such an environment, which may impede the uptake of safe abortion services [40]. As a result, a concerted effort is needed from local administrative and healthcare managers to engage those insecurely housed women in income-generating activities that allow them to access safe abortion and other reproductive and maternal health services. This could also imply that health care providers should offer outreach activities aimed at the most vulnerable segments of the population who are unable to access health care due to financial constraints.

Knowledge of safe abortion services was also identified as a significant determinant of safe abortion services. This finding was supported by studies conducted in Ghana, and northern Ethiopia [25, 41]. This could be attributed to a high likelihood of utilizing safe abortion services due to knowledge of safe abortion services, such as the legal framework of abortion at the national level and the place where safe abortion services are provided.

Finally, the study revealed that those insecurely housed women who could afford the cost of abortion services had a better chance of having a safe abortion than those who couldn't. One of the barriers to safe abortion service use for insecurely housed women was the costs spent for an abortion before and at a health facility, which may encourage them to seek traditional providers even though the service is free at public health facilities [14]. This finding is consistent with the study conducted in the Democratic Republic of Congo (DCR) which indicated that high costs inhibit access to safe abortion [21]. As a result, health care providers in the township needed to place a strong emphasis on raising awareness about the provision of safe abortion services without financial restraints.

The study focuses on the most vulnerable and neglected segments of the population, for whom knowledge on sexual and reproductive health issues is scarce, and the findings are important in developing essential policies, strategies, and programs, as well as planning interventions for these groups. Furthermore, unlike the majority of studies, the current study was conducted among women who have had abortions, which may provide a more accurate picture of safe abortion services. Because the overall number of insecurely housed women with abortion experience in the study areas was not statistically documented, probability sampling procedures were not applied, which could raise the generalizability issue. Even though respondents were given as much time as they needed for a good recall of long-term memory, inquiries were made following an ordered sequence of events; starting with the present and thinking back to a point the likelihood of recall bias should be considered. The study relied on self-reports, and the sensitive nature of the issue may have contributed to social desirability biases.

## Conclusion

The magnitude of safe abortion service utilization among insecurely housed women in the study area was low. Average daily income, knowledge of respondents on safe abortion services, and affordability of the service were identified as significant predictors of safe abortion service utilization. Respective town health offices and health care providers at the facility level should strive to improve awareness about safe abortion services, their legal framework, and their availability. The magnitude of safe abortion service utilization among insecurely housed women in the study area was low. The respective town health offices and health care providers at the facility level should strive to improve awareness about safe abortion service's legal framework, and its availability. In addition, a concerted effort is needed from local administrators, NGOs, and healthcare managers to engage those insecurely housed women in income-generating activities that allow them to access safe abortion and other reproductive and maternal health services.

## Supporting information

**S1 File. The data collection tool used to assess the magnitude of Safe abortion service and its determinants.**
(DOCX)

**S1 Dataset. The minimal data set that supports the findings of the study.**
(SAV)

## Acknowledgments

We are grateful to Jimma University, the faculty of public health, Epidemiology department for providing Ethical approval for this research. For their assistance and support during the study, we thank the managers and healthcare professionals who worked in the selected township. Finally, we want to thank our supervisors, data collectors, and study participants for their contributions throughout the study.

## Author Contributions

**Conceptualization:** Kidist Alemu, Fitsum Endale, Aklilu Habte.

**Data curation:** Aiggan Tamene, Aklilu Habte.

**Formal analysis:** Kidist Alemu, Solomon Birhanu.

**Investigation:** Kidist Alemu, Solomon Birhanu, Leta Fekadu, Fitsum Endale, Aiggan Tamene, Aklilu Habte.

**Methodology:** Kidist Alemu, Solomon Birhanu, Leta Fekadu, Fitsum Endale, Aklilu Habte.

**Software:** Kidist Alemu, Fitsum Endale.

**Supervision:** Kidist Alemu, Solomon Birhanu, Leta Fekadu, Fitsum Endale, Aiggan Tamene.

**Visualization:** Kidist Alemu.

**Writing – original draft:** Aklilu Habte.

**Writing – review & editing:** Kidist Alemu, Solomon Birhanu, Leta Fekadu, Fitsum Endale, Aiggan Tamene, Aklilu Habte.

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
