## [Decision Letter · Decision Letter 0]

6 Jul 2022

PONE-D-21-36728

Safe Abortion Service Utilization and Associated Factors among Homeless Women Who Experienced Abortion in Southwest Ethiopia, 2021: A Community-Based Cross-Sectional Study

PLOS ONE

Dear Dr. Habte,

Thank you for submitting your manuscript to PLOS ONE. After careful consideration, we feel that it has merit but does not fully meet PLOS ONE’s publication criteria as it currently stands. Therefore, we invite you to submit a revised version of the manuscript that addresses the points raised during the review process.

Please note that we have only been able to secure a single reviewer to assess your manuscript. We are issuing a decision on your manuscript at this point to prevent further delays in the evaluation of your manuscript. Please be aware that the editor who handles your revised manuscript might find it necessary to invite additional reviewers to assess this work once the revised manuscript is submitted. However, we will aim to proceed on the basis of this single review if possible.

The manuscript has been evaluated by one reviewer, and his comments are available below.

The reviewer has raised a number of concerns. He requests improvements to the reporting of methodological aspects of the study (such as changes to the terminology used) revisions to the statistical analyses and to the language used.

Could you please carefully revise the manuscript to address all comments raised?

We look forward to receiving your revised manuscript.

Kind regards,

Lorena Verduci

Staff Editor

PLOS ONE

Journal Requirements:

Reviewers' comments:

Reviewer's Responses to Questions

**Comments to the Author**

1. Is the manuscript technically sound, and do the data support the conclusions?

Reviewer #1: Yes

2. Has the statistical analysis been performed appropriately and rigorously? 

Reviewer #1: No

3. Have the authors made all data underlying the findings in their manuscript fully available?

Reviewer #1: Yes

4. Is the manuscript presented in an intelligible fashion and written in standard English?

Reviewer #1: No

5. Review Comments to the Author

Reviewer #1: Thank you for the opportunity to review this manuscript. The manuscript reports the findings of a study of safe-abortion services use among reproductive-age street-involved insecurely housed women in Southwest Ethiopia. The authors explored the prevalence and predictors of accessing safe-abortion services by insecurely housed women by surveying a convenience sample of 124 women and conducting a multivariant logistic regression with use of safe abortion services (yes/no) as the binary outcome variable. The surveyed insecurely housed women were found to be significantly less likely to access safe abortion services than the general population in Ethiopia, while income, knowledge of available safe-abortion services and perceptions about the cost of such services being the significant predictors of not accessing safe-abortions.

The manuscript reports the findings of an important and interesting study on an extremely marginalized and understudied population with important implications for our understanding of reproductive health behaviors and the barriers marginalized populations navigate as they try to access reproductive health services. Most importantly the findings show that the choice is between safe and unsafe abortion services, and that insecurely housed women on the margins of the Ethiopian society, are not utilizing safe and freely available abortion services due to lack of knowledge and misconceptions about costs.

However, the manuscript requires some significant revisions before it will be ready for publication.

1. The manuscript requires careful language editing. At times the inconsistent language, grammatical errors, and some (not many) typos prevents from fully understanding the authors’ intentions.

2. The terminology used is not consistent with current conventions of writing about people experiencing insecure housing and homelessness. Though the authors refer to their sample as homeless women, at least 26% of them were street-involved in that they were panhandling (the authors term this ‘begging’) however they slept indoors. The authors (p.13) differentiate between on-the street and off-the street women with ‘off-the-street” women defined as those sleeping in the street. This may be a typo and the authors meant that the other way around, in which case 70% of the sample were street-involved but insecurely housed rather than homeless. I would suggest terming the sample insecurely housed women rather than homeless women. The terms ‘street girls’ and ‘street women’ (p.10) should be replaced with street-involved

3. The quantitative analysis requires significant revisions. Rather than a proper statistical power analysis the authors used a thumb rule (10 participants – not ‘samples’ as the authors write – per predictor variable) this should be replaced by a post-hoc power analysis reporting the power provided by the 124 participants when running a multivariate logistic regression with 3 predictors (the final model). Reliability of measures is not well reported, Cronbach’s alphas for each scale used should be reported both for the sample and from previous validation studies if available as well as some details about who developed these measures and why these were chosen. The authors state (p.12) they have committed some questions due to low reliability score – the full details of the process need to be provided including which questions were omitted, which scale thy belonged to originally and what were the scores. The process by which the multivariate logistic regression model was developed is not well described, how were the original predictors chosen and how were some of them omitted and the rest retained? The Hosmer-Lemeshow test of goodness of fit reported is usually not recommended due to low reliability. Providing a more fulsome description of the logistic regression conducted and the model fit indices is needed.

4. It is not clear why knowledge about safe-abortion services and the misconception about their cost are distinct factors – surely thinking a free service is unaffordable constitute an example for lack of knowledge – a better explanation as to why this were deemed distinct is needed.

5. The association between higher daily income and access to safe abortion services found is more perplexing than the discussion accounts for. Given the services are provided free of charge, is higher daily income a proxy for some other characteristic (e.g., higher executive functioning; greater autonomy in decision making)?

In sum I wish to reiterate that the manuscript is interesting and the study highly important and I hope some of my suggestions will prove helpful in revising this manuscript and resubmitting for publication.

6. PLOS authors have the option to publish the peer review history of their article (what does this mean?). If published, this will include your full peer review and any attached files.

Reviewer #1: **Yes: **Jonathan Alschech

---

## [Author Response · Author response to Decision Letter 0]

13 Jul 2022

Attached as "Response to Reviewers" in the submission system

---

## [Editor Report · Decision Letter 1]

1 Aug 2022

Safe abortion service utilization and associated factors among Insecurely housed women who experienced abortion in Southwest Ethiopia, 2021: A community-based cross-sectional study

PONE-D-21-36728R1

Dear Dr. Habte,

We’re pleased to inform you that your manuscript has been judged scientifically suitable for publication and will be formally accepted for publication once it meets all outstanding technical requirements.

Kind regards,

Dylan A Mordaunt, MD, MPH, FRACP

Academic Editor

PLOS ONE

Additional Editor Comments (optional):

Thank you for your resubmission. This is a simple but interesting and an important piece of work. The reviewer previously identified a need for major revisions. I've subsequently taken over editing this, but I can see that these issues have been addressed.

With regards to the criteria for publication:

1. The study appears to present the results of original research.

2. Results reported do not appear to have been published elsewhere.

3. Experiments, statistics, and other analyses are performed to a high technical standard and are described in sufficient detail.

4. Conclusions are presented in an appropriate fashion and are supported by the data.

5. The article is presented in an intelligible fashion and is written in standard English.

6. The research meets all applicable standards for the ethics of experimentation and research integrity.

7. The article adheres to appropriate reporting guidelines and community standards for data availability.

Furthermore, following internal discussion on the article, we have decided that the single reviewer secured for the mansucript was able to provide a thorough evaluation of the study and the addition of another reviewer would have yield minimal incremental value.

Although the study may have benefited from an additional qualitative approach, to provide an exploratory element given the research question. The quantitative approach adopted by the authors is acceptable, as it yielded  numerical data. Numerical data may be beneficial for some policy questions where hard numbers are more useful than painting exploratory pictures.

Furthermore the result should also be interpreted with the limitations of the  statistical analysis, given issues in the sub-group comparisons. These issues arise due to a poor overall understanding of what the whole population cohort is characteristics, as a result of both the problem area but specifically related to the sampling approach.

Overall the mansucript satisfies PLOS ONE’s publication criteria and we have decided that it is suitable for publication
---

## [Editor Report · Acceptance letter]

2 Aug 2022

PONE-D-21-36728R1 

Safe abortion service utilization and associated factors among Insecurely housed women who experienced abortion in Southwest Ethiopia, 2021: A community-based cross-sectional study 

Dear Dr. Habte:

I'm pleased to inform you that your manuscript has been deemed suitable for publication in PLOS ONE. Congratulations! Your manuscript is now with our production department. 

Kind regards, 

on behalf of

Associate Professor Dylan A Mordaunt 

Academic Editor

PLOS ONE